# Prevalence, genotype distribution and mutations of hepatitis B virus and the associated risk factors among pregnant women residing in the northern shores of Persian Gulf, Iran

**Reza Taherkhani, Fatemeh Farshadpour** [ID] *

Department of Virology, School of Medicine, Bushehr University of Medical Sciences, Bushehr, Iran

☯ These authors contributed equally to this work.
* f.farshadpour@bpums.ac.ir

**Data Availability Statement:** All relevant data are within the paper.

## Abstract

### Background

Considering perinatal transmission and the high rate of chronic hepatitis B virus (HBV) infection in infants, diagnosis of HBV infection during pregnancy and timely interventions are of great importance. Therefore, this study was performed to investigate the prevalence and genotype distribution of HBV infection and the associated risk factors among pregnant women in the northern shores of the Persian Gulf, South of Iran.

### Methods

Serum samples of 1425 pregnant women were tested for the presence of HBsAg and HBcAb by ELISA (HBsAg one—Version ULTRA and HBc Ab ELISA kits, DIA.PRO, Milan, Italy). The seropositive samples were tested for the presence of HBV DNA by nested PCR, targeting S, X, pre-core (pre-C), and basal core promoter (BCP) regions of the HBV genome. The amplified fragments were sequenced by Sanger dideoxy sequencing technology to evaluate the genotype distribution and mutations of HBV infection by using the MEGA 7 software. The HBV seropositive pregnant women were tested for HCV and HIV coinfections by ELISA (HCV Ab and HIV Ab/Ag ELISA kits, DIA.PRO, Milan, Italy).

### Results

Of 1425 participants, 15 pregnant women (1.05%, 95% CI: 0.64%-1.73%) were positive for HBsAg, 41 women (2.88%, 95% CI: 2.10%-3.88%) were positive for HBcAb, and 5 women (0.35%, 95% CI: 0.15% –0.82%) had HBV viremia with genotype D, sub-genotype D3 and subtype ayw2. One of the viremic samples was positive for HBcAb but negative for HBsAg, which is indicative of an occult HBV infection. HBsAg seroprevalence was higher among pregnant women aged 20 to 29 years, women in the third trimester of pregnancy, residents of Khormuj city, Afghan immigrants, illiterate women, and pregnant women with a history of

**Funding:** This study was funded by Bushehr University of Medical Sciences with grant number 357. The funders had no role in study design, data collection and analysis, decision to publish, or preparation of the manuscript.

**Competing interests:** The authors have declared that no competing interests exist.

tattoo and HBV vaccination. The highest rate of HBcAb seroprevalence was observed in residents of Borazjan city, Turk ethnicity, the age group >39 years, and those women with more parities and a history of abortion. Nevertheless, HBV seroprevalence among pregnant women was not statistically associated with these variables. In contrast, HBcAb seropositivity was significantly associated with the history of tattoo ($P$ = 0.018). According to mutations analyses, seven amino acid substitutions in the HBsAg, one point mutation in the pre-C region, and five points mutations in the BCP region were detected. Besides, the BCP mutations caused amino acid substitutions in the X protein. Of note, the conversion of Ala → Val at amino acid 168 (A168V) and Thr → Pro at amino acid 127 (T127P) were detected in HBsAg of the occult HBV strain.

## Conclusion

These results indicate a relatively low prevalence of HBV infection among pregnant women in the South of Iran, while tattooing is a risk factor for exposure to HBV infection. Moreover, all of the HBV-positive pregnant women were asymptomatic and unaware of their infection. Therefore, routine screening for HBV markers during pregnancy, appropriate treatment of HBV-infected women, and HBV vaccination are recommended to decrease mother-to-child transmission of HBV.

## Introduction

Hepatitis B virus (HBV), a member of the family *Hepadnaviridae*, is characterized by asymptomatic, acute, or chronic hepatitis, which might progress to fulminant hepatic failure, cirrhosis, or hepatocellular carcinoma [1, 2]. Currently, about 250 million of the world population are infected with chronic hepatitis B, and 65 million of those infected are women of childbearing age. HBV-related liver diseases are responsible for approximately 800 000 deaths annually [3].

The HBV genome is a circular partially double-stranded DNA with four overlapping open reading frames: S ORF for surface proteins (HBsAg), P ORF for polymerase (reverse transcriptase), preC/*C* ORF for e antigen (HBeAg) and core protein (HBcAg), and X ORF for X protein [4–6]. Due to the lack of proofreading function of HBV reverse transcriptase, a high rate of mutation naturally occurs during the replication of the viral genome. The most common mutations are pre-core (pre-C) and basal core promoter (BCP) mutations in the HBV core gene [5, 7]. These mutations affect the expression of HBV e antigen (HBeAg) protein and have a significant association with progressive liver diseases such as liver cirrhosis and hepatocellular carcinoma (HCC) [2, 8]. In addition, some mutations in the S region have an important role in the occurrence of occult HBV infection with undetectable levels of HBsAg [9].

Horizontal and perinatal transmissions are the main routes of HBV infection. Horizontal transmission might happen through blood transfusion, intravenous drug use, surgery, tattooing, and sexual intercourse [10, 11]. Perinatal or mother-to-child transmission is one of the most common modes of HBV transmission worldwide. Perinatal transmission of HBV most often occurs during birth and delivery [12, 13]. The risk of HBV infection in children born to mothers who are HBsAg-positive and HBeAg-positive is 70% to 90%, and over 90% of these children eventually develop chronic HBV infection, with approximately 25% progressing to cirrhosis and hepatocellular carcinoma [13–15]. This high rate of chronic HBV infection

indicates a worse prognosis in infants. However, diagnosis of HBV infection during pregnancy or before delivery and timely interventions can mitigate the risk of HBV transmission to fetus or newborn and, consequently, reduce the disease burden in the community [3, 16].

Iran, with a prevalence rate of 3% in the general population, is considered one of the intermediate endemic countries for HBV infection [17]. The initiation of HBV vaccination of infants since 1993 and teenagers since 2006 in the country has had a significant role in reducing the prevalence of HBV infection in the community [18]. Considering the high rates of chronic HBV carriers among those patients who acquire HBV infection perinatally from infected mothers, perinatal transmission is probably the dominant route of HBV transmission in Iran [19]. Therefore, the prevention of maternal infection and its transmission to the fetuses or newborns is essential, so the diagnosis of infected pregnant women is of particular importance. Nevertheless, no report is available on the prevalence of HBV infection among the pregnant population in South of Iran, and despite the profound effects of HBV genotypes and mutants on the progression and treatment of HBV infection, little is known about the molecular analysis of HBV infection in Iran. Therefore, this study was performed to investigate the prevalence, risk factors, and genotype distribution of HBV infection and associated mutations in the S, X, pre-C, and BCP regions of the HBV genome among pregnant women resident in the northern shores of the Persian Gulf, the South of Iran.

## Subjects and methods

### Study setting and population

The study was conducted from January 2018 to June 2019 and included 1425 pregnant women from the public health centers in five cities on the northern shores of the Persian Gulf. The public health centers were chosen according to the multi-stage cluster sampling method. In the first stage, Bushehr, Ahram, Borazjan, Jam, and Khormuj cities of this territory were selected randomly. In the second stage, three public health centers from each city were selected randomly, and all the pregnant women attending these centers for routine visits were included consecutively in this study. The leftover serum samples of pregnant women were used for HBV detection. The pregnant women gave written informed consent to participate in the study and use their serum samples for HBV detection. The socio-demographic characteristics, pregnancy information, history of HBV vaccination and the potential risk factors associated with HBV infection were obtained by interviewing each pregnant woman using a questionnaire. Levels of the liver enzymes were obtained from the clinical records of pregnant women at the public health centers. The Ethics Committee of the Bushehr University of Medical Sciences approved this study with reference number IR.BPUMS.REC.1396.111.

### Laboratory methods

**HBV serological testing.** Serum samples of 1425 pregnant women were tested manually for the presence of hepatitis B surface antigen (HBsAg) and hepatitis B core antibody (HBcAb) using commercially available ELISA kits (HBsAg one—Version ULTRA and HBc Ab ELISA kits, DIA.PRO, Milan, Italy) according to the manufacturer's instructions. The specificity and sensitivity of these kits were 100%. The HBV seropositive pregnant women were tested for the presence of HBeAg as well as HCV and HIV coinfections by ELISA kits (HBe Ag&Ab, HCV Ab, and HIV Ab/Ag ELISA kits, DIA.PRO, Milan, Italy).

**PCR amplification and sequencing.** The seropositive samples were tested to evaluate the genotype distribution and mutations of HBV infection by two nested PCR assays, targeting S, X, pre-C, and BCP regions of the HBV genome [20–24]. Following the extraction of HBV DNA using the High Pure Viral Nucleic Acid kit (Roche, Mannheim, Germany), part of the S

**Table 1. Sequences of primers for amplification of S, X, and pre-core regions of the HBV genome.**

| Virus | Primers Name | Sequences of Primers 5′→3′ | Gene | Region in Genome | Annealing temperature | Size | References |
|---|---|---|---|---|---|---|---|
| HBV | 244-HBS-F1 | GAGTCTAGACTCGTGGTGGACTTC | S | 244–267 | 56°C | 447 bp | [20, 21] |
| | 691-HBS-R1 | AAATKGCACTAGTAAACTGAGCCA | | 668–691 | | | |
| | 255-HBS-F2 | CGTGGTGGACTTCTCTCAATTTTC | | 255–278 | 56°C | 416 bp | |
| | 671-HBS-R2 | GCCARGAGAAACGGRCTGAGGCCC | | 648–671 | | | |
| HBV | 1606-pre-C-F1 | GCATGGAGACCACCGTGAAC | X and pre-core region | 1606–1625 | 57°C | 789 bp | [22–24] |
| | 2395-pre-C-R1 | AGGCGAGGGAGTTCTTCTTC | | 2376–2395 | | | |
| | 1653-pre-C-F2 | CATAAGAGGACTCTTGGACT | | 1653–1672 | 55°C | 740 bp | |
| | 2393-pre-C-R2 | GCGAGGGAGTTCTTCTTC | | 2376–2393 | | | |

region was amplified in the first round PCR using primers 244-HBS-F1 (`GAGTCTAGACTC GTGGTGGACTTC`) and 691-HBS-R1 (`AAATKGCACTAGTAAACTGAGCCA`). The second-round PCR was performed using primers 255-HBS-F2 (`CGTGGTGGACTTCTCTCAATTTTC`) and 671-HBS-R2 (`GCCARGAGAAACGGRCTGAGGCCC`). The 416 bp length fragments from the S region were sequenced by Sanger dideoxy sequencing technology to determine HBV genotypes (Macrogen Co., Korea).

To determine mutations in BCP and pre-C regions, 789 nucleotides and 740 nucleotides of the X and pre-C region were amplified by nested PCR using outer primers [forward primer (1606-pre-C-F1): `GCATGGAGACCACCGTGAAC`; reverse primer (2395-pre-C-R1): `AGGCGA GGGAGTTCTTCTTC`] and inner primers [forward primer (1653-pre-C-F2): `CATAAGAGGACT CTTGGACT`; reverse primer (2393-pre-C-R2): `GCGAGGGAGTTCTTCTTC`], respectively. The 740 bp length fragments from the X and pre-C region were sequenced to determine HBV mutations. The sequences of primers and regions in the HBV genome for detection of HBV genotypes and mutants are summarized in Table 1.

**Phylogenetic and mutational analysis.** The obtained sequences from the S, X, and pre-C regions of the HBV genome were aligned and compared with the reference sequences representing the standard HBV genotypes available at the nucleotide database of the NCBI by ClustalW program in the Molecular Evolutionary Genetics Analysis (MEGA) software version 7.0 (Biodesign Institute, Tempe, AZ, USA). The phylogenetic tree was constructed by the neighbor-joining method, as described previously [25]. Moreover, the mutations of S, pre-C, BCP, and X regions of the HBV genome isolated from pregnant women were determined. The S, X, and pre-C sequences isolated from pregnant women were submitted to the GenBank sequence database.

## Statistical analysis

The $\chi^2$ test or Fisher's exact test was used to analyze and compare categorical variables between HBV-positive and HBV-negative pregnant women. Quantitative variables were compared by the Student's $t$-test. Univariate and multivariate logistic regression analyses were performed to evaluate the risk factors associated with HBV infection among pregnant women. The data were analyzed by SPSS 17 package program (SPSS Inc., Chicago, IL, USA), and $P$ values $< 0.05$ were defined as statistically significant.

## Results

Of 1425 pregnant women, 616 women were from Bushehr city, 207 women were from Ahram city, 40 women were from Khormuj city, 440 women were from Borazjan city and 122 women

were from Jam city. The mean age ± SD of pregnant women was 28.1±5.99 years with a range of 14–46 years. The majority of pregnant women were in the third trimester of pregnancy (68.4%), the age group 20–29 years (55.2%), educated (97.1%), and Fars (90.0%). Of 1425 participants, 15 pregnant women (1.05%, 95% CI: 0.64%-1.73%) were positive for HBsAg, and 41 women (2.88%, 95% CI: 2.10%-3.88%) had HBcAb.

HBsAg seropositive pregnant women had a lower mean age (27.53±5.96) compared to HBsAg seronegative pregnant women (28.11±5.99), while this difference was not statistically significant (*P* = 0.71). Moreover, there was no difference between the mean age of HBcAb seropositive pregnant women (28.88±5.96) and HBcAb seronegative pregnant women (28.08 ±5.99) (*P* = 0.402). The majority of HBsAg seropositive women were in the age group 20–29 years (1.4%), the third trimester of pregnancy (1.3%), residents in Khormuj city (2.5%), Afghan (1.6%), illiterate (2.4%) and had a history of tattoo (1.6%) and HBV vaccination (1.6%) (S1 Table). The highest rate of HBcAb seroprevalence was observed in residents of Borazjan city (3.9%), Turk and Afghan immigrants (8.3% and 6.5%, respectively), the age group >39 years (4.3%), illiterate pregnant women (7.1%), pregnant women in the second trimester of pregnancy (3.6%), and those women with more parities (5.0%) and a history of abortion (3.0%) (S2 Table). However, none of these variables were statistically associated with HBsAg and HBcAb seroprevalence. In contrast, HBcAb seropositivity was significantly associated with the history of tattoo, so that tattooing was a risk factor for exposure to HBV infection (OR: 1.47; 95% CI: 1.06–2.04; *P* = 0.018) (S2 Table). In addition, HBcAb seropositivity was more prevalent in those samples collected in October (11.5%) compared to the other months. However, according to the multivariate logistic regression analysis, the difference in the monthly rate of HBcAb seroprevalence was statistically insignificant. All of the HBV seropositive samples had normal levels of liver enzymes and were negative for HCV and HIV. The prevalence of HBsAg and HBcAb among pregnant women grouped according to socio-demographic characteristics and qualitative variables are presented in S1 and S2 Tables, respectively.

Of 1425 pregnant women, 5 women (0.35%, 95% CI: 0.15%– 0.82%) had HBV viremia with genotype D, sub-genotype D3, and subtype ayw2 (GenBank accession Nos. OK490614-OK490616) (Fig 1). One of the viremic samples was positive for HBcAb but negative for HBsAg, which is indicative of an occult HBV infection. Overall, 26.7% of HBsAg seropositive pregnant women (4 of 15 HBsAg positive women) and 2.7% of HBcAb seropositive women (1 of 37 HBsAg negative and HBcAb positive women) had HBV viremia. Notably, 3 samples were positive in the first round of nested PCR, and 2 samples were found to be positive in the second round of nested PCR (Figs 2 and 3). Seven amino acid substitutions were detected in HBsAg of strains isolated from pregnant women, including the conversion of Ile → Leu at amino acid 110 (I110L), Thr → Pro at amino acid 127 (T127P), Asp → Glu at amino acid 144 (D144E), Ser → Ala at amino acid 154 (S154A), Gly → Ala at amino acid 159 (G159A), Ala → Gly at amino acid 166 (A166G) and Ala → Val at amino acid 168 (A168V) of HBsAg (S1 Fig and Table 2). Of note, A168V and T127P were detected in the HBsAg of the occult HBV strain (HB51). Moreover, a G to A substitution at nucleotide position 1896 (G1896A) was detected in the pre-C region of one sample (HB55) (S2 Fig), this sample was negative for HBeAg. T1753A, A1761C, G1764T/A, C1766G/T, and C1773T mutations were detected in the BCP region. Of these, T1753A and A1761C mutations were detected in one sample (HB55); and G1764T/A, C1766G/T, and C1773T mutations were detected in two samples (HB35 and HB55). T1753A, A1761C, and G1764A mutations caused the conversion of Ile → Asn at amino acid 127 (I127N), Lys → Gln at amino acid 130 (K130Q), and Val → Leu at amino acid 131 (V131L) of the X protein, respectively (S3 Fig).

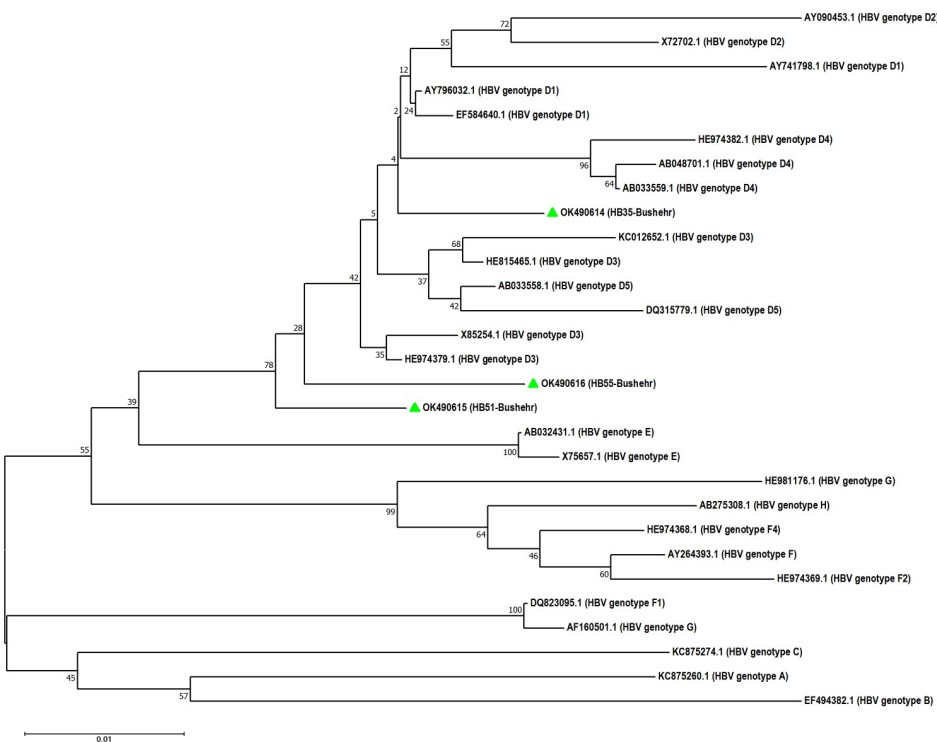

**Fig 1. Neighbor-joining phylogenetic tree based on ~380 bp nucleotide sequence (~270–670 bp of the complete reference genome) of S region of HBV isolates (green mark) from the serum samples of pregnant women in South of Iran (GenBank accession Nos. OK490614-OK490616).** Bootstrap resampling strategy and reconstruction were carried out 1,000 times to confirm the reliability of the phylogenetic tree.

## Discussion

HBV causes chronic infections in 2% to 3% of adults and 95% of infants and is an important factor in the development of advanced chronic liver disease and HCC in these patients [3]. Notably, perinatal or mother-to-infant transmission is responsible for the occurrence of over 50% of chronic HBV carriers [13]. Considering the high rates of perinatal transmission and

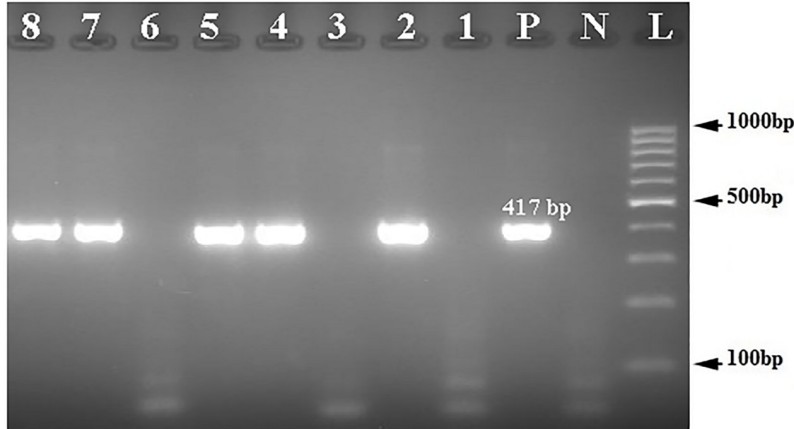

**Fig 2. Electrophoresis of PCR products of S region of HBV genome extracted from serum samples of pregnant women on 2% agarose gel.** L, 100-bp DNA ladder; N, negative control; P, positive control; 1, 2, 3, 4, 6, 7 and 9, amplified products (≈417 bp).

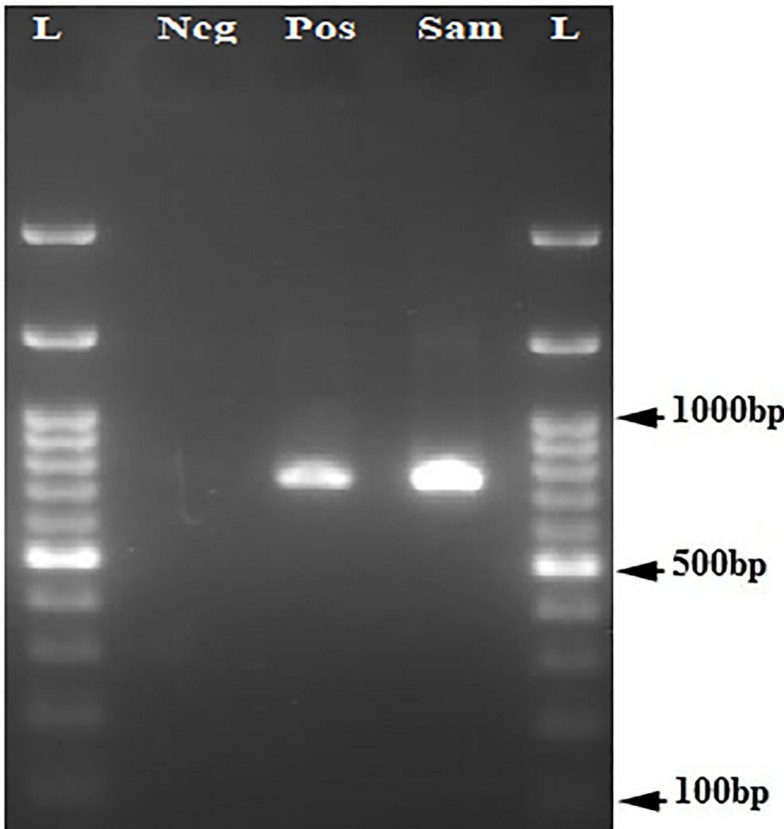

**Fig 3. Electrophoresis of PCR products of X and pre-core regions of HBV genome extracted from serum samples of pregnant women on 2% agarose gel.** L, 100-bp DNA ladder; Neg, negative control; Pos, positive control; Sam, amplified product (≈740 bp).

chronic HBV infection in infants, diagnosis of HBV infection among pregnant women is of great importance. Despite this importance, there is no report on the prevalence of HBV infection among pregnant women in the northern shores of the Persian Gulf, South of Iran. Therefore, we evaluated the prevalence and molecular epidemiology of HBV infection among pregnant women in this region and found HBV prevalence of 1.05% for HBsAg, 2.88% for HBcAb, and 0.35% for HBV viremia with genotype D, sub-genotype D3, and subtype ayw2.

**Table 2. Frequency of point replacement mutations in HBsAg of strains isolated from pregnant women in the South of Iran.**

| Position S mutation | Frequency of point mutation No. (%) | Patient No. |
|---|---|---|
| I110L | 1 (33%) | HB55 |
| T127P | 2 (66%) | HB51, HB55 |
| D144E | 1 (33%) | HB35 |
| S154A | 1 (33%) | HB55 |
| G159A | 1 (33%) | HB55 |
| A166G | 1 (33%) | HB35 |
| A168V | 2 (66%) | HB51, HB55 |

The HBsAg seroprevalence of 1.05% observed in pregnant women is considerably higher than the HBsAg prevalence of 0.15% reported in the blood donors of this region [18]. Besides, the HBV prevalence noted in this study is slightly lower than the overall prevalence of HBV among Iranian pregnant women (1.18%) [15]. In this study, all of the HBV seropositive pregnant women were asymptomatic and unaware of their infection. These infected but asymptomatic pregnant women may remain undiagnosed over time. The risk of HBV infection in children born to infected mothers is 70% to 90%, and over 90% of these children eventually develop chronic HBV infection, with approximately 25% progression to cirrhosis and HCC [13–15]. This high rate of chronic HBV infection indicates a worse prognosis in infants. Routine screening for HBV infection during pregnancy is therefore recommended for the identification of HBV-infected pregnant women and neonates born at risk. Maternal treatment with oral antivirals such as tenofovir, telbivudine, and lamivudine and immunoglobulin administration at birth can prevent mother-to-child transmission of HBV. Consequently, this reduces the burden of chronic HBV infection in the community [3, 16]. According to the results of this study, 0.35% of pregnant women had HBV viremia. Prenatal transmission, which is leading to postpartum immunoprophylaxis incompetence, is possible in women with HBV viremia [14]. Since a significant percentage of pregnant women were not vaccinated, HBV vaccination is recommended to reduce the incidence of HBV infection during pregnancy.

The prevalence of 1.05% for HBsAg observed in the present study is higher than those observed among pregnant women in other parts of Iran, 0.18% in Babol (North of Iran) [26], 0.3% in Mashhad (North-East of Iran) [27], 0.59% in Dehloran (West of Iran) [28], 0.7% in Lorestan (West of Iran) [29], 0.77% in Kurdistan (West of Iran) [30] and 0.87% in East Azerbaijan (North-West of Iran) [30] but lower than 1.3% in Rafsanjan (South-East of Iran) [31], 1.93% in Golestan (North of Iran) [30], 2.26% in Jiroft (South-East of Iran) [30] and 2.5% in Malekan (North-West of Iran) [32]. The HBsAg prevalence noted in this study is also higher than those observed among pregnant women of some other countries, 0.9% in Brazil [33] but lower than those reported among pregnant women in Ghana (3.3%) [34], Central Nigeria (5.5%) [35], Shenyang, China (5.5%) [36], Ethiopia (7.9%) [37], Gambia (9.2%) [38], North Cameroon (10.2%) [39] and Yemen (10.8%) [40]. Moreover, the prevalence of 2.88% for HBcAb reported in this study is higher than that observed among pregnant women in Malekan (1.25%) [32] but lower than that in Lorestan (3.4%) [29]. This prevalence is lower than those of other countries such as Brazil (7.4%) [33] and Shenyang, China (29.65%) [36]. These variations in the seroprevalence of HBV are probably because of differences in rates of high-risk behaviors, level of awareness, immunization status, the routes of transmission, epidemiology of HBV, and burden of the infection in different regions. However, differences in sociodemographic characteristics of the participants and the sensitivity and specificity of the diagnosis assays in different studies can also explain these variations.

In this study, HBV seroprevalence among the pregnant women was not statistically associated with age, stage of gestation, number of pregnancies, place of residency, level of education, ethnicity, and history of blood transfusion, abortion, dentistry, surgery, and HBV vaccination, although the majority of HBsAg seropositive pregnant women were in the age group 20–29 years, the third trimester of pregnancy, residents in Khormuj city, illiterate, Afghan and had a history of tattoo and HBV vaccination. Furthermore, the highest rate of HBcAb seroprevalence was observed in residents of Borazjan city, Turk and Afghan immigrants, the age group >39 years, illiterate pregnant women, pregnant women in the second trimester of pregnancy, and those women with more parities and a history of abortion. The higher seroprevalence of HBV among pregnant women resident in Khormuj and Borazjan cities might be due to the higher prevalence of HBV infection in the general population of these cities. In contrast, HBcAb seropositivity was significantly associated with the time of sampling and the history of tattoo, so

that tattooing was a risk factor for exposure to HBV infection. In addition, HBcAb seropositivity was more prevalent in those samples collected in October compared to the other months. The reason for this monthly pattern of HBcAb seroprevalence is not clear.

A recent meta-analysis study revealed that HBsAg seroprevalence among pregnant women in Iran is associated with the history of blood transfusion, abortion, illiteracy, and intravenous drug use by spouses [15]. In a study from the South-East of Iran, HBsAg seroprevalence among pregnant women was associated with illicit drug use and tattooing [31]. In another study from the West of Iran, more parities and abortion were risk factors for exposure to HBV [28]. In a recent study from Yemen, a significant association was found between female circumcision and HBsAg seropositivity [40]. Another study from North Cameroon reported a significant association between HBV infection and a history of blood transfusion [39]. Previous studies from Ethiopia demonstrated that HBsAg seroprevalence among pregnant women is associated with multiple sexual partners, history of abortion, dentistry, and surgery [37, 41]. This discrepancy between reports might be due to differences in the predominant routes of transmission and risk behavior patterns in different geographical regions.

In the present study, the highest rate of HBsAg seroprevalence was observed in pregnant women aged 20 to 29 years (1.4%). Considering the compulsory vaccination of infants since 1993 and teenagers since 2006 [18], this is probably due to the partial ineffectiveness or incompleteness of the national HBV vaccination program in this age group. However, further studies are required to determine the efficacy of hepatitis B vaccination among pregnant women in this region. Moreover, there was no difference between HBV-vaccinated and HBV-unvaccinated pregnant women regarding the prevalence of HBsAg. Therefore, although there are effective tools and strategies for the prevention and treatment of HBV infection, management of HBV infection during pregnancy faces challenges that need further investigation. On the other hand, all pregnant women under 20 years old were negative for HBsAg, which is probably due to the initiation of neonatal vaccination in the country in 1993, awareness of HBV infection among youths, and less exposure to HBV or less engagement in risky behaviors in young ages. In contrast, a recent study in Central Nigeria demonstrated a higher prevalence of HBsAg among pregnant women under 20 years old [35].

In this study, occult HBV infection was detected in one sample with undetectable levels of HBsAg and low HBV DNA levels. Nevertheless, P120T and G145R mutations associated with failure of HBsAg detection were not found in this sample. P120T and G145R mutations in the S gene are associated with low responses to HBV vaccination and immunoglobulin therapy as well as failures of diagnostic tests [42, 43]. In this study, one point mutation in the pre-C region and five points mutations in the BCP region were detected. Of these, the pre-C mutation (G1896A) eliminates HBeAg expression at the translational level, whereas BCP mutations (G1764A) decrease HBeAg expression at the transcriptional level [5]. HBeAg is an immune target, and lack of its expression may result in loss of immune responses and subsequently liver disease progression [7, 44]. Notably, the sample with G1896A mutation was HBeAg negative. The G1896A mutation causes the conversion of TGG→TAG (Trp → Stop codon) at codon 28 of the pre-C gene and the suppression of HBeAg expression. Whereas, the BCP mutations enhance the replication rate of the HBV genome [5]. The BCP and pre-C mutations are associated with the progression of chronic HBV infection to advanced liver disease and are frequently found in patients with chronic hepatitis, fibrosis, liver cirrhosis, and HCC [2, 7]. Therefore, early detection of these mutations is pivotal in predicting the clinical progression in patients with chronic HBV infection and deciding the antiviral medications. No nucleotide insertions, deletions, or frameshifting were observed in our samples.

The dominance of genotype D is the most important characteristic of HBV infection among our pregnant women, which reflects the genotypic pattern of HBV infection in our

region. Genotype D is also the predominant genotype in Iran [45]. To date, 10 HBV genotypes (A to J) with 9 serological subtypes and over 35 sub-genotypes related to genotypes A–D, F, H, and I have been identified to be not only predictive of response to antiviral treatment and disease progression but also associated with modes of transmission and geographical regions [6, 10]. Of them, genotype D is the main genotype in Africa, Europe, India, Indonesia, and the Mediterranean countries, and is characterized by chronicity, worse clinical outcomes (cirrhosis and HCC), low response to IFN-based therapy, and a high frequency of BCP and pre-C mutations [10, 11]. The pre-C G1896A mutation is frequently found in HBV genotypes B, D, and E [44]. Moreover, the BCP A1762T and G1764A mutations are reported in association with HBV genotypes C and D [46]. On the other hand, sub-genotype D3 is prevalent among injecting drug users [47]. Like HCV infection [48], injecting drug use is most probably the main route of HBV transmission in this region. However, we were unable to confirm this claim, since the majority of women were not willing to answer the questions regarding the intravenous drug use by themselves or more importantly by their spouses.

This is the first report on the prevalence of HBV infection among pregnant women resident in the northern shores of the Persian Gulf, and the first to investigate the seroprevalence of HBsAg and HBcAb among pregnant Afghan migrant women in Iran. Moreover, the results of this study can be generalized to the pregnant population of this territory due to the consecutive recruitment of participants and the high number of assessed pregnant women. However, due to the cross-sectional design of the study, we were not able to determine the eventual effects of hepatitis B on pregnancy outcomes and the infants born to infected mothers. Therefore, this report should be completed by prospective studies. Besides, due to the lack of awareness of the level of immunity against hepatitis B among the study population, this study was not able to determine the efficacy of hepatitis B vaccination among pregnant women in this region. Nevertheless, antibody to hepatitis B surface antigen (anti-HBsAg) was not assessed among the study population. Therefore, there is a need to study the effectiveness of the immunization policy in the pregnant population of this region.

In this study, the nested-PCR assay was used to detect HBV infection among pregnant women. This two-step PCR method has a very high sensitivity for the diagnosis of cases with low virus levels. However, in comparison with quantitative PCR (qPCR), the nested-PCR assay is not able to determine the viral loads. Moreover, the main limitation of nested-PCR is the probability of contamination that can cause false-positive results. In this study, strict quality control was applied, and positive serum samples were re-tested to ensure the absence of contamination and the accuracy of the results.

## Conclusion

The results of this study indicate a relatively low prevalence of HBV infection among pregnant women in the South of Iran, while tattooing is a risk factor for exposure to HBV infection. Moreover, all of the HBV-positive pregnant women were asymptomatic and unaware of their infection. Therefore, routine screening for HBV markers during pregnancy, appropriate treatment and HBV vaccination of HBV-infected women are recommended to decrease mother-to-child transmission of HBV and, consequently, reduce the disease burden in the community. Moreover, we observed occult HBV infection in one sample with an undetectable level of HBsAg. Therefore, screening of pregnant women based on HBsAg detection is not reliable for the identification of occult HBV infection. The use of more sensitive molecular techniques such as nested-PCR assay is recommended to detect HBV DNA and increase the chance of diagnosing occult hepatitis B with low virus titers. Besides, several point replacement mutations were detected in the S, X, pre-C, and BCP regions of HBV isolated from pregnant

women. Mutation analysis of HBV can be useful in predicting the disease progression in HBV-infected patients.

## Supporting information

**S1 Fig. Alignment of the amino acid sequences of HBsAg (44–170 aa) of strains isolated from pregnant women (GenBank accession Nos. OK490614-OK490616) and the reference sequences available at the GenBank.**
(TIF)

**S2 Fig. Alignment of the amino acid sequences of pre-core region isolated from pregnant women (GenBank accession Nos. OK490617 and OK490618) and the reference sequences available at the GenBank.**
(TIF)

**S3 Fig. Alignment of the amino acid sequences (109–154 aa) of X protein (1698–1838 bp nucleotide sequence) isolated from pregnant women (GenBank accession Nos. OK490617 and OK490618) and the reference sequences available at the GenBank.**
(TIF)

**S1 Table. Prevalence of HBsAg according to socio-demographic and qualitative variables among pregnant women in the South of Iran.**
(DOC)

**S2 Table. Prevalence of HBcAb according to socio-demographic and qualitative variables among pregnant women in the South of Iran.**
(DOC)

**S1 Questionnaire. Research questionnaire.**
(DOC)

**S1 Raw images.**
(PDF)

## Acknowledgments

The authors would like to thank the Deputy Research and Affairs of the Bushehr University of Medical Sciences, Bushehr, Iran.

## Author Contributions

**Formal analysis:** Reza Taherkhani.

**Investigation:** Reza Taherkhani, Fatemeh Farshadpour.

**Methodology:** Reza Taherkhani, Fatemeh Farshadpour.

**Project administration:** Reza Taherkhani, Fatemeh Farshadpour.

**Supervision:** Fatemeh Farshadpour.

**Validation:** Reza Taherkhani.

**Writing – original draft:** Fatemeh Farshadpour.

**Writing – review & editing:** Fatemeh Farshadpour.

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
