## [Decision Letter · Decision Letter 0]

23 Nov 2021

PONE-D-21-32739Prevalence, genotype distribution and mutations of hepatitis B virus infection and associated risk factors among pregnant women resident in the northern shores of Persian Gulf, IranPLOS ONE

Dear Dr. Fatemeh Farshadpour

Thank you for submitting your manuscript to PLOS ONE. After careful consideration, we feel that it has merit but does not fully meet PLOS ONE’s publication criteria as it currently stands. Therefore, we invite you to submit a revised version of the manuscript that addresses the points raised during the review process.

We look forward to receiving your revised manuscript.

Kind regards,

Maemu Petronella Gededzha, Ph.D

Academic Editor

PLOS ONE

Journal Requirements:

“The authors would like to acknowledge grant number supported by the Deputy Research and 393 Affairs of the Bushehr University of Medical Sciences, Bushehr, Iran.”

“This study was funded by Bushehr University of Medical Sciences with grant number 357.

Additional Editor Comments:

The manuscript in the present form demands a **“major revision”** before it can be published. Therefore, we invite you to submit a revised version of the manuscript that addresses the points raised during the review process.

1. Please ensure that your manuscript meets PLOS ONE's style requirements, including those for headings. The PLOS ONE style templates can be found at

2. Quality of written English is not suitable for publication unless extensively edited eg. The word “uneducated” should be replaced by “Illiterate”.

3. The limitations of the study are not sufficiently discussed.

4. Abstract needs to be re-written to reflect the study more correctly eg. Amplification of X gene is not included.

Minor comments

Change laboratory diagnosis sub-heading to laboratory methods.

The author should consider using subheadings in the "laboratory diagnosis” section for easier reading.

Suggestion: HBV serological testing, PCR amplification and sequencing, phylogenetic and mutational analysis or sequence analysis.

Include the full name of the ELISA kit used.

Table 1- Include degrees (0C) in the annealing temperature column.

The author indicate that the sequences has been submitted to GenBank, but no accession numbers provided.

The study relied on nucleotide sequence analyses. However, there is no mention of sequencing protocol/approach/platform used in generating the nucleotide sequences in the methods section, only referenced a company that looks like it provides service.

Table 2 and 3 should be moved to supplementary.

One of the city is stated as Khomorj in Table 2 and throughout it is stated as Khomurj.

Phylogenetic and mutational analysis (Figure 3 and Table 4) focused on 3 samples yet they detected HBV DNA in 5 samples. It is not clear whether the other 2 samples could not be sequenced or why they were left out of the analysis.

Figure 3: Only indicate bootstrap values >70% which is considered moderate confidence

Reviewers' comments:

Reviewer's Responses to Questions

**Comments to the Author**

1. Is the manuscript technically sound, and do the data support the conclusions?

Reviewer #1: No

Reviewer #2: Yes

Reviewer #3: Yes

Reviewer #4: Yes

2. Has the statistical analysis been performed appropriately and rigorously? 

Reviewer #1: No

Reviewer #2: Yes

Reviewer #3: Yes

Reviewer #4: Yes

3. Have the authors made all data underlying the findings in their manuscript fully available?

Reviewer #1: Yes

Reviewer #2: Yes

Reviewer #3: Yes

Reviewer #4: Yes

4. Is the manuscript presented in an intelligible fashion and written in standard English?

Reviewer #1: No

Reviewer #2: Yes

Reviewer #3: Yes

Reviewer #4: Yes

5. Review Comments to the Author

Reviewer #1: The author used unacceptable language (e.g. uneducated women) which also feels derogatory. The manuscript used unnecessary statistics, while its main research was HBV infection (HBsAg or HBV DNA) or exposure as determined by HBcAb. Only 15 pregnant women were HBsAg positive but the manuscript made a lots of unnecessary and unacceptable statistical analysis (smoking, Education to mention few).

Reviewer #2: This was a well designed study that showed the prevalence and molecular characterization of HBV in pregnant woman. The literature review clearly stated the problem and the aim of the study. The methodology used was standard for this type of study. In the results section, there is a comment on the occult status in Line 193, it is not shown what the viral load of this sample is in order to make such a statement. I suggest the author includes viral load of this sample. In the discussion, in Line 255, the author states that the HBV prevalence of 1.05% was lower than 1.18%, these are similar especially if rounded off. In line 258, Risk of mother to child transmission should be discussed more here, they did not show what the risk of undiagnosed HBV in pregnancy would be, especially in those infected in the third trimester. Line 268, This recommendation is not clear. Do they mean that they will test each and every women in the South of Iran area? this is not cost effective, rather screen all women who come in for their first visit at the ANC and vaccinate the baby at birth. Or treat the infection if it becomes chronic. Line 319, Cannot make this conclusion as this was not part of the questions asked in the questionnaire as shown in the information on Table 2 and 3. You only have vaccination data for 14 participants out of over 1000. This is being very presumptuous and this conclusion needs to be made with caution. Line 326, this statement contradicts what the authors had said in Line 318-320 on the ineffectiveness of the HBV vaccination programme. Line 331, What was the viral load of this sample? it was also not mentioned in the results section. Line 339-340, show the importance of this lack of HBeAg during pregnancy, this has not been discussed fully here. The Conclusion on Line 376 differs from the conclusion on the abstract. The one in the abstract lines up with what the authors set out to do, while the one in Line 376 is a repetition of what the results stated.

Reviewer #3: Title: suggestion, change “resident” to “residing”

Abstract: Page 2, line 32 please mention the name of the ELISA kit used, company and the sentence that follows should read tested for the presence of HPV DNA delete “detection”.

Page 2, line 34 the software or platforms used to analyse the mutations are not mentioned

Page 2, line 43, the sentence talking about tattooing being a risk factor should come in the conclusion. Or the sentence can either be stated as significantly associated or as risk factor not two of them.

Page 2, line 36, did the HBcAb positive women have HBsAg or were they only positive for HBcAb? This is because in your results you state that 2.7% of HBcAb positive were also positive for HBV DNA.

Page ,2 line 46 Please indicate how many had which genotype.

Page 2, line 49, in the methods the authors mention the pre-core regions however in the results they mention pre-core and basal core regions.

Page 2, line 48, it is important to note that the mutations were detected in the same sample deemed as an occult, it is not clear in the statement.

Page 2, line 50 to 53, the authors are not concluding on their findings but are making a general remark. First it should be clear if HBV is endemic or not in this region and if perinatal transmission is a problem or not. In their results the authors found a very low prevalence, they should say something about it in their conclusion.

Page 4, line 95, please add the information about the target genes missing in the abstract.

Page 7, line 167, means 1.6% of the vaccinated women had chronic or acute infection.

Page 8, line 177, the authors did not mention, any liver enzyme tests performed however they are mentioning them in their results.

Page 8, line 177-179, the tables should also be referred to in the text.

Page 11, line 190, a suggestion to delete “evaluation” to “testing”

Page 11, line 190 the authors should stick to one decimal after the comma throughout the document for consistency.

Page 14, line 258 to 263 the sentence is too long,

Why did the authors not check the women vaccination status? Why did the authors not test for Anti-HBs but the authors are making a statement in the discussion that majority of women are not vaccinated in this region. Do the authors mean to tell us that there is no HBV routine testing in pregnant women in this region?

Page 16, line 319-320 the authors cannot claim that the HBV vaccination programs are not effective while they did not report if these people had received the vaccine or not and check why if they did not received the vaccine?

Page 17, line 323, how did the authors arrive at a conclusion that the HBV management during pregnancy is facing challenges?

Page 17, line 332 please rephrase the sentence

This sample was found to be positive in the second round of PCR; this statement is irrelevant in the discussion.

Page 17, line 340, the authors state that one sample was negative for HBeAg however this was not mentioned in the methods that was tested in any sample.

Page 19, line 374, the statement is not clear.

General comments

The authors should try to keep their sentences short.

The pictures are not clear; they are not of good quality.

The authors in their conclusion make no mention of the risk factors associated with HBV infection in their study.

The authors can summarize the results section in the abstract.

Information on vaccination in the study area is not mentioned in the introduction, the vaccination coverage, when did the vaccination starts, which age group is targeted, the HBV testing program for pregnant women in this area is also missing in the introduction. These informs the reader about the situation in the study area and supports why study was conducted.

There is no made mention of how the sample size was calculated only how the study sites were identified.

Reviewer #4: While the findings of this study will fill a critical knowledge gap and contribute to informing interventions aimed at preventing HBV mother-to-child transmission in Iran, there is need for further clarity in the reporting of the methodology followed and results obtained in order to improve the quality of the manuscript. In addition, the limitations of the study are not explicitly addressed in the manuscript.

6. PLOS authors have the option to publish the peer review history of their article (what does this mean?). If published, this will include your full peer review and any attached files.

Reviewer #1: **Yes: **Azwidowi Lukhwareni

Reviewer #2: No

Reviewer #3: No

Reviewer #4: **Yes: **Edina Amponsah-Dacosta

---

## [Author Response · Author response to Decision Letter 0]

17 Dec 2021

ANSWERING REVIEWERS

December 15, 2021

Dear Editor-in-Chief of

PLOS ONE

We would greatly appreciate for your consideration of the publication of our manuscript entitled "Prevalence, genotype distribution and mutations of hepatitis B virus infection and associated risk factors among pregnant women resident in the northern shores of Persian Gulf, Iran" in the PLOS ONE.

Ms. No. PONE-D-21-32739

Title: Prevalence, genotype distribution and mutations of hepatitis B virus infection and associated risk factors among pregnant women resident in the northern shores of Persian Gulf, Iran

Name of Journal: PLOS ONE

Revision has been made according to the suggestions of the reviewers and the editor.

We greatly appreciated the reviewers’ comments. The point-to-point responses to comments were shown as follows:

Journal Requirements:

and

Response: The PLOS ONE's style requirements were applied.

“The authors would like to acknowledge grant number supported by the Deputy Research and 393 Affairs of the Bushehr University of Medical Sciences, Bushehr, Iran.”

“This study was funded by Bushehr University of Medical Sciences with grant number 357.

Response: The funding-related texts were removed from the Methods and the Acknowledgment sections of the manuscript. The funding information present in the Funding Statement section of the online submission form is correct.

Response: The original uncropped and unadjusted images underlying gel results were provided. In the original uncropped figure 1, seven positive samples are shown. Since, two positive samples (positive samples number 4 and 5) were loaded again in the wells number 12 and 13. The reason was we did not want to have empty wells, we tried to load all of the wells. 

Response: The ethics statement was removed from end of the manuscript.

Response: Captions for the Supporting Information files were included at the end of the manuscript.

Additional Editor Comments:

The manuscript in the present form demands a “major revision” before it can be published. Therefore, we invite you to submit a revised version of the manuscript that addresses the points raised during the review process.

Response: We greatly appreciate you taking the time to consider our manuscript while providing us constructive comments. Moreover, thank you very much for giving us the opportunity to answer the reviewers’ comments. We are really grateful for this opportunity.

1. Please ensure that your manuscript meets PLOS ONE's style requirements, including those for headings. The PLOS ONE style templates can be found at

Response: The PLOS ONE's style requirements were applied.

2. Quality of written English is not suitable for publication unless extensively edited eg. The word “uneducated” should be replaced by “Illiterate”.

Response: The manuscript was edited for proper English language, grammar, punctuation, and spelling by our team at Bushehr University of Medical Sciences.

3. The limitations of the study are not sufficiently discussed. 

Response: The limitations of the study were discussed in more details and highlighted.

4. Abstract needs to be re-written to reflect the study more correctly eg. Amplification of X gene is not included. 

Response: The abstract was re-written.

Minor comments:

Change laboratory diagnosis sub-heading to laboratory methods. 

Response: The change was done.

The author should consider using subheadings in the "laboratory diagnosis” section for easier reading. 

Suggestion: HBV serological testing, PCR amplification and sequencing, phylogenetic and mutational analysis or sequence analysis. 

Response: The suggestions were applied.

Include the full name of the ELISA kit used. 

Response: The full name of the ELISA kits was included and highlighted with green color.

Table 1- Include degrees (℃) in the annealing temperature column. 

Response: The degrees (℃) in the annealing temperature column was included.

The author indicate that the sequences has been submitted to GenBank, but no accession numbers provided.

Response: The accession numbers were provided and highlighted with yellow color in the result section and the legends of Fig3 and S1-S3 Figs.

The sequences have been submitted to GenBank and the accession numbers were provided via email, but they have not been released yet.

The study relied on nucleotide sequence analyses. However, there is no mention of sequencing protocol/approach/platform used in generating the nucleotide sequences in the methods section, only referenced a company that looks like it provides service. 

Response: The Sanger dideoxy sequencing technology was included in the method section.

Table 2 and 3 should be moved to supplementary. 

Response: The table 2 and 3 were moved to supplementary. 

One of the city is stated as Khormoj in Table 2 and throughout it is stated as Khormuj. 

Response: The spelling was corrected in the table 2.

Phylogenetic and mutational analysis (Figure 3 and Table 4) focused on 3 samples yet they detected HBV DNA in 5 samples. It is not clear whether the other 2 samples could not be sequenced or why they were left out of the analysis.

Response: Unfortunately, the other 2 samples had low-quality sequencing results, so that it was impossible to include them in the phylogenetic and mutational analysis.

Figure 3: Only indicate bootstrap values >70% which is considered moderate confidence.

Response: The phylogenetic tree was constructed by the neighbor-joining method with the Bootstrap test of phylogeny (substitutional model: maximum composite likelihood) in Molecular Evolutionary Genetics Analysis (MEGA) program version 7. Bootstrap resampling strategy and reconstruction were carried out 1,000 times to confirm the reliability of the phylogenetic tree.

Reviewer's Responses to Questions

Comments to the Author

1. Is the manuscript technically sound, and do the data support the conclusions?

Reviewer #1: No

Reviewer #2: Yes

Reviewer #3: Yes

Reviewer #4: Yes

2. Has the statistical analysis been performed appropriately and rigorously? 

Reviewer #1: No

Reviewer #2: Yes

Reviewer #3: Yes

Reviewer #4: Yes

3. Have the authors made all data underlying the findings in their manuscript fully available?

Reviewer #1: Yes

Reviewer #2: Yes

Reviewer #3: Yes

Reviewer #4: Yes

4. Is the manuscript presented in an intelligible fashion and written in standard English?

Reviewer #1: No

Reviewer #2: Yes

Reviewer #3: Yes

Reviewer #4: Yes

5. Review Comments to the Author

Reviewer #1: The author used unacceptable language (e.g. uneducated women) which also feels derogatory. The manuscript used unnecessary statistics, while its main research was HBV infection (HBsAg or HBV DNA) or exposure as determined by HBcAb. Only 15 pregnant women were HBsAg positive but the manuscript made a lot of unnecessary and unacceptable statistical analysis (smoking, Education to mention few).

Response: Thank you very much for your review. 

-The manuscript was edited for proper English language, grammar, punctuation, and spelling.

-These variables were included in the study in order to find the possible risk factors of HBV infection among pregnant women.

Regarding smoking: In addition to cigarette, some women in the South of Iran use hookah for smoking and they share the hookah in group. We thought that there might be a chance of HBV transmission through saliva by using hookah.

Regarding education: The education level has an indirect effect on the prevalence of HBV. The educated women have more information about the transmission routes of HBV.

Regarding the month of sampling: We assumed that there might be a seasonal or monthly pattern regarding the seroprevalence of HBV infection among pregnant women in this region.

According to those studies on the seasonality of hepatitis, we wanted to investigate the seasonal pattern in this study too.

Reviewer #2: This was a well-designed study that showed the prevalence and molecular characterization of HBV in pregnant woman. The literature review clearly stated the problem and the aim of the study. The methodology used was standard for this type of study. 

Response: Thank you very much for your review and the positive comments. 

In the results section, there is a comment on the occult status in Line 193, it is not shown what the viral load of this sample is in order to make such a statement. I suggest the author includes viral load of this sample.

Response: Occult HBV infection is described by the presence of HBV DNA in the absence of detectable HBsAg in serum or plasma. This sample, despite being positive for HBV DNA, was negative for HBsAg, which is indicative of an occult HBV infection.

-We did not measure HBV viral loads since the nested PCR assay used in this study is a qualitative method and is not able to determine the viral loads. This is a limitation of nested-PCR over quantitative PCR (qPCR). The limitations of nested PCR assay were included in the manuscript and highlighted with yellow color in the limitations section of the study.

-Those samples with very low virus levels become positive in the second round of nested PCR. This sample with occult HBV infection was identified in the second round of amplification, which is indicative of low HBV DNA levels. 

- In one of our previous research projects (unpublished data), the viral load of one of the HBV DNA positive serum samples was measured by real time PCR, and the serum sample was serial diluted. The diluted sample with 25 HBV DNA copies/ml (5 HBV DNA copies/200 μL) became positive in the second round of nested PCR, while it was negative in the first round of PCR.

In the discussion, in Line 255, the author states that the HBV prevalence of 1.05% was lower than 1.18%, these are similar especially if rounded off.

Response: We cannot round off the numbers here. Because regarding the prevalence rate, even a small decimal is important. We used “slightly” in the sentence.

In line 258, Risk of mother to child transmission should be discussed more here, they did not show what the risk of undiagnosed HBV in pregnancy would be, especially in those infected in the third trimester. 

Response: The mentioned issue was included and highlighted with yellow color.

Line 268, This recommendation is not clear. Do they mean that they will test each and every women in the South of Iran area? this is not cost effective, rather screen all women who come in for their first visit at the ANC and vaccinate the baby at birth. Or treat the infection if it becomes chronic. 

Response: We agree with the dear reviewer. This sentence was revised and highlighted with green color.

Line 319, Cannot make this conclusion as this was not part of the questions asked in the questionnaire as shown in the information on Table 2 and 3. You only have vaccination data for 14 participants out of over 1000. This is being very presumptuous and this conclusion needs to be made with caution.

Response: We agree with the dear reviewer. Therefore, the sentence was revised and highlighted with green color.

-A significant percentage of the pregnant women in this study were not vaccinated according to self-declaration during the interview. Only 22.0% of the participants were vaccinated, 22.0% were not vaccinated and 55.9% of the participants did not respond to the questions regarding history of HBV vaccination during interview (Unknown group). These participants did not know anything about HBV vaccination. Considering the compulsory vaccination of infants since 1993 and teenagers since 2006 in Iran, individuals above 30 years were not the target group of national vaccination program. Since a high number of the “Unknown group” were above 30 years, we think that a significant percentage of the pregnant women in this study were not vaccinated. These data are shown in the tables 2 and 3.

-According to the aim of the study and the available funding, the presence of hepatitis B surface antigen (HBsAg), hepatitis B core antibody (HBcAb) and HBV viremia was investigated in serum samples of the participants. Unfortunately, we did not assess antibody to hepatitis B surface antigen (anti-HBsAg) among the study population due to lack of budget. This issue was mentioned as the limitation of the study, and it is suggested to study the effectiveness of the immunization policy in the pregnant population of this region.

Line 326, this statement contradicts what the authors had said in Line 318-320 on the ineffectiveness of the HBV vaccination program. 

Response: The previous statement has been revised. 

-The previous statement was about the seroprevalence of HBV in the age group 20-29 years old. This statement is about the pregnant women under 20 years old.

Line 331, What was the viral load of this sample? it was also not mentioned in the results section. 

Response: We did not measure HBV viral loads since the nested PCR assay used in this study is a qualitative method and is not able to determine the viral loads. This a limitation of nested-PCR over qPCR. The limitations of nested PCR assay were included in the manuscript.

Those samples with very low virus levels become positive in the second round of nested PCR. This sample with occult HBV infection was identified in the second round of amplification, which is indicative of low HBV DNA levels. 

-In one of our previous research projects (unpublished data), the viral load of one of the HBV DNA positive serum samples was measured by real time PCR, and the serum sample was serial diluted.

The diluted serum sample with 25 HBV DNA copies/ml (5 HBV DNA copies/200 μL) became positive in the second round of nested PCR, while it was negative in the first round of PCR.

Line 339-340, show the importance of this lack of HBeAg during pregnancy, this has not been discussed fully here. 

Response: No, this sentence does not show the importance of the lack of HBeAg during pregnancy. This shows the relationship between the expression of HBeAg and immune responses to HBV infection.

It shows that the absence of HBeAg may result in evasion of immune clearance as HBeAg is a target antigen for immune recognition.

The sample with G1896A mutation was HBeAg negative. The G1896A mutation causes the conversion of TGG�TAG (Trp � Stop codon) at codon 28 of the pre-C gene and the suppression of HBeAg expression. This HBeAg negative sample is at the risk of liver disease progression.

In fact, it wants to show that the mutation analysis of HBV infection can be useful in predicting the disease progression in HBV-infected patients.

The Conclusion on Line 376 differs from the conclusion on the abstract. The one in the abstract lines up with what the authors set out to do, while the one in Line 376 is a repetition of what the results stated. 

Response: The conclusion was revised and highlighted.

Reviewer #3: Title: suggestion, change “resident” to “residing” 

Response: Thank you very much for your review and the constructive comments. 

- The suggestion was applied and highlighted with yellow color.

Abstract: Page 2, line 32 please mention the name of the ELISA kit used, company and the sentence that follows should read tested for the presence of HBV DNA delete “detection”. 

Response: The full name of the ELISA kits was included.

Page 2, line 34 the software or platforms used to analyze the mutations are not mentioned. 

Response: The software was included.

Page 2, line 43, the sentence talking about tattooing being a risk factor should come in the conclusion. Or the sentence can either be stated as significantly associated or as risk factor not two of them.

Response: The abstract was revised.

Page 2, line 36, did the HBcAb positive women have HBsAg or were they only positive for HBcAb? This is because in your results you state that 2.7% of HBcAb positive were also positive for HBV DNA.

Response: Of 41 HBcAb seropositive women, 4 women were positive for HBsAg (HBsAg positive and HBcAb positive), and 37 women were negative for HBsAg (HBsAg negative and HBcAb positive). 

Overall, 26.7% of HBsAg seropositive pregnant women and 2.7% of HBcAb seropositive women had HBV viremia with genotype D, sub-genotype D3, and subtype ayw2:

2.7% (1 case of 37 HBsAg negative and HBcAb positive cases)

26.7% (4 cases of 15 HBsAg positive cases)

Page ,2 line 46 Please indicate how many had which genotype.

Response: All of the HBV DNA positive samples had HBV viremia with genotype D, sub-genotype D3 and subtype ayw2.

Page 2, line 49, in the methods the authors mention the pre-core regions however in the results they mention pre-core and basal core regions. 

Response: The target genes were included.

Page 2, line 48, it is important to note that the mutations were detected in the same sample deemed as an occult, it is not clear in the statement.

Response: The mutations were not specific to one sample. The mutations were detected in the S, X, BCP and pre-C regions of the HBV genome isolated from pregnant women and were not specific to the occult HBV sample. 

According to mutations analyses, seven amino acid substitutions in the HBsAg, one point mutation in the pre-C region and five points mutations in the basal core promoter (BCP) region were detected. Besides, the BCP mutations caused amino acid substitutions in the X protein.

Page 2, line 50 to 53, the authors are not concluding on their findings but are making a general remark. First it should be clear if HBV is endemic or not in this region and if perinatal transmission is a problem or not. In their results the authors found a very low prevalence, they should say something about it in their conclusion. 

Response: The conclusion was revised.

Page 4, line 95, please add the information about the target genes missing in the abstract. 

Response: The target genes were included.

Page 7, line 167, means 1.6% of the vaccinated women had chronic or acute infection.

Response: Yes, it means 1.6% of the vaccinated women were HBsAg seropositive, or the seroprevalence of HBsAg among vaccinated women was 1.6%.

Page 8, line 177, the authors did not mention, any liver enzyme tests performed however they are mentioning them in their results.

Response: Levels of the liver enzymes were obtained from the records of pregnant women at the public health centers. The pregnant women attend these public health centers for periodical checkups.

Page 8, line 177-179, the tables should also be referred to in the text. 

Response: The tables were referred to in the text.

Page 11, line 190, a suggestion to delete “evaluation” to “testing” 

Response: The suggestion was applied.

Page 11, line 190 the authors should stick to one decimal after the comma throughout the document for consistency. 

Response: We cannot round off the numbers here. Because regarding the prevalence rate, even a small decimal is important.

Page 14, line 258 to 263 the sentence is too long.

Response: The long sentence was rephrased.

Why did the authors not check the women vaccination status? Why did the authors not test for Anti-HBs but the authors are making a statement in the discussion that majority of women are not vaccinated in this region? Do the authors mean to tell us that there is no HBV routine testing in pregnant women in this region?

Response: A significant percentage of the pregnant women in this study were not vaccinated according to self-declaration during the interview. Only 22.0% of the participants were vaccinated, 22.0% were not vaccinated and 55.9% of the participants did not respond to the questions regarding history of HBV vaccination during interview (Unknown group). These participants did not know anything about HBV vaccination. Considering the compulsory vaccination of infants since 1993 and teenagers since 2006 in Iran, individuals above 30 years were not the target group of national vaccination program. Since a high number of the “Unknown group” were above 30 years, we think that a significant percentage of the pregnant women in this study were not vaccinated. These data are shown in the tables 2 and 3.

-According to the aim of the study and the available funding, the presence of hepatitis B surface antigen (HBsAg), hepatitis B core antibody (HBcAb) and HBV viremia was investigated in serum samples of the participants. Unfortunately, we did not assess antibody to hepatitis B surface antigen (anti-HBsAg) among the study population due to lack of budget. However, we agree with the dear reviewer that the lack of awareness of the level of immunity among the study population can influence interpretation of the study findings. This issue was mentioned as the limitation of the study, and it is suggested to study the effectiveness of the immunization policy in the pregnant population of this region.

-Yes, there is no guideline or national policy on HBV screening during pregnancy. In fact, this study was performed to convince the health care providers to include HBV screening in the routine screening of pregnant women in Iran. Currently, pregnant women are only screened for rubella and CMV in Iran. 

Page 16, line 319-320 the authors cannot claim that the HBV vaccination programs are not effective while they did not report if these people had received the vaccine or not and check why if they did not received the vaccine?

Response: We agree with the dear reviewer. Therefore, the sentence was revised and highlighted with green color.

- This issue was mentioned as the limitation of the study, and it is suggested to study the effectiveness of the immunization policy in the pregnant population of this region.

Page 17, line 323, how did the authors arrive at a conclusion that the HBV management during pregnancy is facing challenges?

Response: Because although there are HBV vaccine and oral antivirals and screening tools for the prevention, treatment and diagnosis of HBV infection, the status of immunity and prevalence of HBV infection among pregnant population in Iran is unknown. Since, there is no guideline or national policy on HBV screening during pregnancy. We need such kind of studies among pregnant women to convince the health care providers to include HBV screening in the management program of pregnant women in Iran.

Page 17, line 332 please rephrase the sentence. 

Response: The sentence was rephrased and highlighted with yellow color.

This sample was found to be positive in the second round of PCR; this statement is irrelevant in the discussion. 

Response: The sentence was removed.

Page 17, line 340, the authors state that one sample was negative for HBeAg however this was not mentioned in the methods that was tested in any sample.

Response: Unfortunately, the presence of HBeAg was not investigated in the serum samples of all participants due to the lack of budget (the available funding could not cover the expenses to test all of the samples). Only the seropositive samples were tested for the presence of HBeAg. Therefore, detection of HBeAg was not mentioned in the methods section. 

Page 19, line 374, the statement is not clear.

Response: Due to the cross-sectional design of the study, we could not determine the possible effects of hepatitis B on pregnancy outcomes. Therefore, prospective or longitudinal studies are required to determine the possible effects of hepatitis B on pregnancy outcomes among pregnant women in the South of Iran.

Prospective studies are required to complete our study regarding the possible effects of hepatitis B on pregnancy outcomes.

General comments

The authors should try to keep their sentences short. 

Response: The long sentences were rephrased.

The pictures are not clear; they are not of good quality. 

Response: The pictures with high quality were provided.

The authors in their conclusion make no mention of the risk factors associated with HBV infection in their study. 

Response: The conclusion was revised.

The authors can summarize the results section in the abstract. 

Response: The abstract was revised.

Information on vaccination in the study area is not mentioned in the introduction, the vaccination coverage, when did the vaccination starts, which age group is targeted, the HBV testing program for pregnant women in this area is also missing in the introduction. These informs the reader about the situation in the study area and supports why study was conducted.

Response: The information on vaccination is mentioned in the introduction and highlighted with yellow color.

-There is no national policy on HBV screening during pregnancy in Iran. In fact, this study was performed to convince the health care providers to include HBV screening in the routine screening of pregnant women in Iran. Currently, pregnant women are only screened for rubella and CMV in Iran. 

There is no made mention of how the sample size was calculated only how the study sites were identified.

Response: This is a cross-sectional study that was performed from January 2018 to June 2019. 

During the study period, all pregnant women who agreed to participate and gave written informed consent to use their samples for HBV detection and their data for analysis were included consecutively in this study.

-This has been mentioned in the methods section: “From January 2018 to June 2019, all the pregnant women attending these public health centers for routine visits were included consecutively in this study. The pregnant women gave written informed consent to participate in the study and use their serum samples for HBV detection.”

Reviewer #4: While the findings of this study will fill a critical knowledge gap and contribute to informing interventions aimed at preventing HBV mother-to-child transmission in Iran, there is need for further clarity in the reporting of the methodology followed and results obtained in order to improve the quality of the manuscript. In addition, the limitations of the study are not explicitly addressed in the manuscript.

Response: Thank you very much for your review and the positive comments. The mentioned issues were included and highlighted.

Reviewer’s Comments

Given the increased lifetime risk for potentially fatal chronic liver disease following perinatally acquired hepatitis B virus (HBV) infection, this study sought to characterize the burden of HBV infection among pregnant women living in the South of Iran. While the findings of this study will fill a critical knowledge gap and contribute to informing interventions aimed at preventing HBV mother-to-child transmission in Iran, there is need for further clarity in the reporting of the methodology followed and results obtained in order to improve the quality of the manuscript.

Response: Thank you very much for your review and the positive comments. 

Major Revisions:

The introduction section does not address universal hepatitis B vaccination globally and in Iran, and how this has influenced the burden of disease over time. In addition, the authors do not address the availability, accessibility and utilization of maternal HBV screening and management programmes in Iran. These should be addressed to provide further clarity to the context of the study. 

Response: The mentioned issue was included and highlighted with yellow color in the introduction section.

-Unfortunately, there is no guideline or national policy on HBV screening during pregnancy. In fact, this study was performed to convince the health care providers to include HBV screening in the routine screening of pregnant women in Iran. Currently, pregnant women are only screened for rubella and CMV in Iran. 

In the methods section of the manuscript, the authors adequately address how socio-demographic, and clinical data were collected from study participants. However, there is no information on how blood samples were collected for the study.

Response: The leftover serum samples of pregnant women who attend the public health centers for routine checkups were used for HBV detection. 

The authors allude to the use of a questionnaire in collecting relevant participant data. For validation purposes, kindly indicate of this questionnaire was piloted. Was a previously published questionnaire used during this study (in which case it should be appropriately cited and referenced) or if a study specific one was developed, can a template be provided as part of supplementary material. 

Response: The socio-demographic characteristics and pregnancy information were obtained by interviewing each pregnant woman using a questionnaire (a study specific one was developed). The questionnaire was translated to English and provided as supplementary.

The authors have not addressed why antibodies to the hepatitis B surface antigen (anti-HBs) was not tested among the study population. It will be interesting to understand from the introduction section how long Iran has had a universal hepatitis B vaccination programme and whether any of the younger participants should have been vaccinated in infancy. A lack of awareness of the level of immunity (either due to vaccination or recovery from a past infection) among the study population may influence interpretation of the study findings and should therefore be adequately addressed.

Response: According to the aim of the study and the available funding, the presence of hepatitis B surface antigen (HBsAg), hepatitis B core antibody (HBcAb) and HBV viremia was investigated in serum samples of the participants. Unfortunately, we did not assess antibody to hepatitis B surface antigen (anti-HBsAg) among the study population due to lack of budget. However, we agree with the dear reviewer that the lack of awareness of the level of immunity among the study population can influence interpretation of the study findings. This issue was mentioned as the limitation of the study, and it is suggested to study the effectiveness of the immunization policy in the pregnant population of this region.

The limitations of the study are not explicitly addressed in the manuscript. 

Response: The limitations of the study were discussed in more details and highlighted.

Minor Revisions: 

Line 61, page 3; “HBV-related liver diseases are responsible for approximately 8,00,000 deaths…” Consider revising the number format (800 000) for better clarity. 

Response: The number format was revised.

Lines 116 – 117, page 5; “…using commercially available ELISA kits (DIA.PRO, Milan, Italy).” For better clarity and reliability, the authors should briefly provide further information (e.g., type of platform used, automated or manual, etc) about the ELISA used. In addition, did the authors follow the manufacturer’s instructions or a published protocol (cite appropriately). 

Response: The mentioned issues were included and highlighted with green color.

Line 118, page 5; “…distribution and mutations of HBV infection by two nested PCR assays, targeting S, X and pre-C…” Was this an in-house PCR assay or did the authors follow a previously published protocol – in which case this should be appropriately cited and referenced. Also, important to note is the fact that the authors have not addressed the limitations of the nested PCR assay used (vs qPCR) in the manuscript.

Response: We used a previously published protocol, and the referenced articles were cited in the table 1.

The strengths and limitations of nested PCR assay were included in the manuscript and highlighted.

Line 148, page 6; “…Logistic regression analysis was used to evaluate the risk factors…” Kindly clarify if univariate and multivariate logistic regression analyses were performed.

Response: Univariate and multivariate logistic regression analyses were performed using HBsAg and HBcAb seroprevalence as dependent variables and socio-demographic characteristics and qualitative variables (age, stage of gestation, number of pregnancies, smoking, place of residency, level of education, ethnicity, time of sampling, and history of blood transfusion, tattoo, abortion, dentistry, surgery and HBV vaccination), as independent variables.

Line 167, page 7; “…uneducated (2.4%) and had a history of tattoo (1.6%) and HBV vaccination (1.6%).” Kindly clarify in the manuscript whether participants’ vaccination history was self-reported only or confirmed using clinical records. The limitations of relying on self-reported clinical history / patient recall, should be adequately addressed in the manuscript where appropriate.

Response: The participants’ vaccination history was obtained according to self-declaration during the interview. This was reported in the method section and highlighted with green color.

-Unfortunately, we did not assess antibody to hepatitis B surface antigen (anti-HBsAg) among the study population due to lack of budget. This issue was mentioned as the limitation of the study and highlighted with green color, and it is suggested to study the effectiveness of the immunization policy in the pregnant population of this region.

Lines 176 – 177, page 8; “All of the HBV seropositive samples had normal levels of liver enzymes and were negative for HCV and HIV.” In the methods section, the authors do not make mention of testing for liver enzymes, HIV or HCV co-infection. The authors should address how they came about these findings reported here, i.e., tested in this study, self-reported during questionnaire / interview, confirmed from reliable clinical records of participants.

Response: Levels of the liver enzymes were obtained from the records of pregnant women at the public health centers. The pregnant women attend these public health centers for periodical checkups. This was mentioned in the method section and highlighted with green color.

- The HBV seropositive pregnant women were tested for HIV and HCV coinfections by ELISA. This issue was included in the method section and highlighted with yellow color.

Lines 257 – 258, page 14; “These infected but asymptomatic pregnant women may remain undiagnosed over time.” As a matter of translating research to practice, can the authors comment on the direct benefits of the study findings to the participants in this case, i.e., were hepatitis B results reported to clinicians managing participants in order to inform timely interventions?

Response: The results of this study were informed to the public health centers and the Deputy Research and Affairs of the Bushehr University of Medical Sciences. Surely, the results have been communicated to the infected pregnant women by the authorities in the public health centers. However, we are not sure about receiving treatment or care post-pregnancy by HBV positive women. Since, the follow-up of infected pregnant women was outside the scope of the study. This issue has been mentioned as a limitation of this cross-sectional study in the manuscript.

Lines 260 – 263, page 14; “Since maternal treatment with oral antivirals…HBV vaccination and immunoglobulin administration at birth…consequently, reduce the burden of chronic HBV infection in the community [3, 16].” The authors should expand on what the national policies and guidelines in relation to these key interventions are, with an indication of the current coverage / utilization rates of these services in the country.

Response: National policy and intervention services for HBV infected pregnant women are the same as reported in the manuscript:

1) Maternal treatment with oral antivirals such as tenofovir, telbivudine and lamivudine

2) HBV vaccination and immunoglobulin administration at birth 

Lines 287 – 288, page 15; “…and the sensitivity and specificity of ELISA kits in different studies can also explain these variations.” The authors should explicitly address the strengths and limitations of assays (nested PCR vs qPCR, not testing for anti-HBs or HBeAg, sensitivity and specificity of ELISA) used in this study compared to those in the aforementioned studies and how this may influence interpretation of the study findings.

Response: The strengths and limitations of nested PCR assay were included in the manuscript and highlighted (limitation section of the study).

- Sensitivity and specificity of the ELISA kits were included in the method section.

Lines 301 – 303, page 16; “In addition, HBcAb seropositivity was more prevalent in those samples collected in October...” It is unclear why the authors assessed the frequency of detection of HBV infection in relation to the months in which samples were collected. Were the authors expecting a seasonal pattern in relation to the prevalence of HBV infection? Have such seasonal patterns been demonstrated elsewhere and if so, what would the relevance of this be?

Response: We did not find a seasonal pattern regarding the seroprevalence of HBV infection among pregnant women in this region. However, according to those studies on the seasonality of hepatitis, we wanted to investigate the monthly or seasonal pattern in this study too.

Lines 309 – 310, page 16; “…a significant association was found between circumcision and HBsAg seropositivity.” With regards to the study cited, could the authors clarify if this implies male circumcision among spouses / sexual partners of pregnant women found to be HBsAg seropositive?

Response: It is female circumcision, which is usually performed at home.

Lines 330 – 331, page 17; “…undetectable levels of HBsAg and low HBV DNA levels.” Did the authors measure HBV viral loads?

Response: No, we did not measure HBV viral loads since the nested PCR assay used in this study is a qualitative method and is not able to determine the viral loads. This a limitation of nested-PCR over qPCR. The limitations of nested PCR assay were included in the manuscript.

-Those samples with very low virus levels become positive in the second round of nested PCR. This sample with occult HBV infection was identified in the second round of amplification, which is indicative of low HBV DNA levels. 

-In one of our previous research projects (unpublished data), the viral load of one of the HBV DNA positive serum samples was measured by real time PCR, and the serum sample was serial diluted. The diluted serum sample with 25 HBV DNA copies/ml (5 HBV DNA copies/200 μL) became positive in the second round of nested PCR, while it was negative in the first round of PCR.

Line 340, page 17; “Notably, the sample with G1896A mutation was HBeAg negative.” In the methods section the authors did not indicate serological testing for additional markers aside for HBsAg and anti-HBc (HBcAB). Could the authors address how they came about this finding in the methods section and why it was not reported as part of the results?

Response: Unfortunately, the presence of HBeAg was not investigated in the serum samples of all participants due to the lack of budget (the available funding could not cover the expenses to test all of the samples). Only the seropositive samples were tested for the presence of HBeAg. Therefore, detection of HBeAg was not mentioned in the methods section.

Lines 349 – 350, page 18; “The dominance of genotype D is the most important characteristic…in our region.” In addition to the stated predominant HBV genotype in the region, the authors should indicate the HBV genotype distribution in Iran as this has not been provided as part of the introduction section.

Response: The HBV genotype distribution in Iran was provided and highlighted with green color.

Sincerely yours,

Fatemeh Farshadpour, Ph.D, 

Bushehr University of Medical Sciences 7514633341 

Bushehr, Iran.

00989171712653

E-mail: f.farshadpour@bpums.ac.ir

f.farshadpour@yahoo.com

---

## [Decision Letter · Decision Letter 1]

2 Feb 2022

PONE-D-21-32739R1Prevalence, genotype distribution and mutations of hepatitis B virus infection and associated risk factors among pregnant women residing in the northern shores of Persian Gulf, IranPLOS ONE

Dear Dr. Farshadpour

Thank you for submitting your manuscript to PLOS ONE. After careful consideration, we feel that it has merit but does not fully meet PLOS ONE’s publication criteria as it currently stands. Therefore, we invite you to submit a revised version of the manuscript that addresses the points raised during the review process.

 1.The manuscript needs an accurate English revision and an accurate format check.2. Please address minor comments requested by reviewer. 3Please ensure that your decision is justified on PLOS ONE’s publication criteria and not, for example, on novelty or perceived impact.

Please submit your revised manuscript by Mar 19 2022 11:59PM.  If you will need more time than this to complete your revisions, please reply to this message or contact the journal office at plosone@plos.org. Please include the following items when submitting your revised manuscript:A rebuttal letter that responds to each point raised by the academic editor and reviewer(s). You should upload this letter as a separate file labeled 'Response to Reviewers'.A marked-up copy of your manuscript that highlights changes made to the original version. You should upload this as a separate file labeled 'Revised Manuscript with Track Changes'.An unmarked version of your revised paper without tracked changes. You should upload this as a separate file labeled 'Manuscript'.If applicable, we recommend that you deposit your laboratory protocols in protocols.io to enhance the reproducibility of your results. Protocols.io assigns your protocol its own identifier (DOI) so that it can be cited independently in the future. For instructions see: https://journals.plos.org/plosone/s/submission-guidelines#loc-laboratory-protocols. Additionally, PLOS ONE offers an option for publishing peer-reviewed Lab Protocol articles, which describe protocols hosted on protocols.io. Read more information on sharing protocols at https://plos.org/protocols?utm_medium=editorial-email&utm_source=authorletters&utm_campaign=protocols.

We look forward to receiving your revised manuscript.

Kind regards,

Maemu Petronella Gededzha, Ph.D

Academic Editor

PLOS ONE

Journal Requirements:

Reviewers' comments:

Reviewer's Responses to Questions

**Comments to the Author**

1. If the authors have adequately addressed your comments raised in a previous round of review and you feel that this manuscript is now acceptable for publication, you may indicate that here to bypass the “Comments to the Author” section, enter your conflict of interest statement in the “Confidential to Editor” section, and submit your "Accept" recommendation.

Reviewer #3: (No Response)

Reviewer #4: All comments have been addressed

2. Is the manuscript technically sound, and do the data support the conclusions?

Reviewer #3: Yes

Reviewer #4: Yes

3. Has the statistical analysis been performed appropriately and rigorously? 

Reviewer #3: Yes

Reviewer #4: Yes

4. Have the authors made all data underlying the findings in their manuscript fully available?

Reviewer #3: Yes

Reviewer #4: Yes

5. Is the manuscript presented in an intelligible fashion and written in standard English?

Reviewer #3: Yes

Reviewer #4: No

6. Review Comments to the Author

Reviewer #3: Dr RL Lebelo review

Thank you to the Authors for addressing the comments and suggestions that I have made to the manuscript. Thank you for improving on your work overall presentation. I have am happy with the paper however I have few comments and further comets on some previous review.

Title: Prevalence, genotype distribution and mutations of hepatitis B virus infection and associated risk factors among pregnant women residing in the northern shores of Persian Gulf, Iran.

On the title the mutations cannot be of infection but of the virus thus the title should be reworded.

I suggest that after ‘and’ before associated the authors add “the”, the same should be done in page 2 line 28.

Page 2, line 48, it is important to note that the mutations were detected in the same sample deemed as an occult, it is not clear in the statement.

11

Response: The mutations were not specific to one sample. The mutations were detected in the S,

X, BCP and pre-C regions of the HBV genome isolated from pregnant women and were not specific to the occult HBV sample.

According to mutations analyses, seven amino acid substitutions in the HBsAg, one point mutation in the pre-C region and five points mutations in the basal core promoter (BCP) region were detected. Besides, the BCP mutations caused amino acid substitutions in the X protein.

Reviewer: the authors still do not show which mutations were found in the sample deemed occult. The statement is not clear meaning all samples showed the same mutations on all genes targeted.

Page 2, line 36, did the HBcAb positive women have HBsAg or were they only positive for

HBcAb? This is because in your results you state that 2.7% of HBcAb positive were also positive for HBV DNA.

Response: Of 41 HBcAb seropositive women, 4 women were positive for HBsAg (HBsAg positive and HBcAb positive), and 37 women were negative for HBsAg (HBsAg negative and HBcAb positive).

Overall, 26.7% of HBsAg seropositive pregnant women and 2.7% of HBcAb seropositive women had HBV viremia with genotype D, sub-genotype D3, and subtype ayw2:

2.7% (1 case of 37 HBsAg negative and HBcAb positive cases)

26.7% (4 cases of 15 HBsAg positive cases)

Reviewer: Page 13 line 227, can the authors add the percentages on these sentences because they are confusing where they are placed. Then the authors can mention the genotypes without the percentages.

In page 6 line 128-129 the authors make mention of the act that seropositive women were tested for HIV and HCV, was this part of this manuscript? Or another study? This is because the results section in the abstract make no mention of these results.

I page 6 line 132 the authors mention two nested PCR assay however the author only make mention of one PCR assay targeting the S gene.

Page 7 line 148-153 falls under the title “PCR amplification and sequencing” from line 153 to 156 should be added to the text under phylogenetic and mutational analysis and rephrased.

Page 8, line 177, the authors did not mention, any liver enzyme tests performed however they are mentioning them in their results.

Response: Levels of the liver enzymes were obtained from the records of pregnant women at the public health centers. The pregnant women attend these public health centers for periodical checkups.

Reviewer: can this statement be added in the methods.

Page 8 line 179 after (68.4%) the sentence is unclear and should be rephrased.

Page 17, line 340, the authors state that one sample was negative for HBeAg however this was not mentioned in the methods that was tested in any sample.

Response: Unfortunately, the presence of HBeAg was not investigated in the serum samples of all participants due to the lack of budget (the available funding could not cover the expenses to test all of the samples). Only the seropositive samples were tested for the presence of HBeAg.

Therefore, detection of HBeAg was not mentioned in the methods section.

Reviewer: it does not matter how many samples were tested but the reason behind the selection of the samples tested should be mentioned otherwise the authors should remove the statement about HBeAg from the results.

Page 16 line 368, the authors can according to literature discuss the outcomes of the children infected with the genotypes detected and the implications of the mutations found if they were to be transmitted to the children. I therefor feel that the authors are discussing the results and not discussing the results looking at the population and the implications of the infections transmitted to the children.

Page 17 line 390, the “of HBV-infected women” should come after HBV vaccination so that the recommended can include vaccination of the infected women not just vaccination.

The last comment is based on vaccinated in Iran yes there is vaccination in infants does the country do a birth dose of vaccine? This should be included in the discussion as a way to reduce the prevalence.

Reviewer #4: The authors have adequately considered the comments made in the preliminary review and revising the manuscript. I propose that the manuscript be reviewed for minor grammatical and typographical errors to enhance clarity and improve readability.

7. PLOS authors have the option to publish the peer review history of their article (what does this mean?). If published, this will include your full peer review and any attached files.

Reviewer #3: **Yes: **Dr Ramokone Lisbeth Lebelo

Reviewer #4: **Yes: **Edina Amponsah-Dacosta

---

## [Author Response · Author response to Decision Letter 1]

7 Feb 2022

ANSWERING REVIEWERS

February 7, 2022

Dear Editor-in-Chief of

PLOS ONE

We would greatly appreciate for your consideration of the publication of our manuscript entitled "Prevalence, genotype distribution and mutations of hepatitis B virus infection and associated risk factors among pregnant women resident in the northern shores of Persian Gulf, Iran" in the PLOS ONE.

Ms. No. PONE-D-21-32739R1

Title: Prevalence, genotype distribution and mutations of hepatitis B virus infection and associated risk factors among pregnant women resident in the northern shores of Persian Gulf, Iran

Name of Journal: PLOS ONE

Revision has been made according to the suggestions of the reviewers.

We greatly appreciated the reviewers’ comments. The point-to-point responses to comments were shown as follows:

Journal Requirements:

Response: The reference list was checked.

Review Comments to the Author

Reviewer #3: Dr RL Lebelo review

Thank you to the Authors for addressing the comments and suggestions that I have made to the manuscript. Thank you for improving on your work overall presentation. I am happy with the paper however I have few comments and further comets on some previous review.

Response: Thank you very much for your review.

Title: Prevalence, genotype distribution and mutations of hepatitis B virus infection and associated risk factors among pregnant women residing in the northern shores of Persian Gulf, Iran.

On the title the mutations cannot be of infection but of the virus thus the title should be reworded.

Response: “infection” was removed.

I suggest that after ‘and’ before associated the authors add “the”, the same should be done in page 2 line 28.

Response: The suggestion was applied.

Page 2, line 48, it is important to note that the mutations were detected in the same sample deemed as an occult, it is not clear in the statement.

Response: The mutations were not specific to one sample. The mutations were detected in the S,

X, BCP and pre-C regions of the HBV genome isolated from pregnant women and were not specific to the occult HBV sample.

According to mutations analyses, seven amino acid substitutions in the HBsAg, one point mutation in the pre-C region and five points mutations in the basal core promoter (BCP) region were detected. Besides, the BCP mutations caused amino acid substitutions in the X protein.

Reviewer: the authors still do not show which mutations were found in the sample deemed occult. The statement is not clear meaning all samples showed the same mutations on all genes targeted.

Response: No, all samples did not show the same mutations on all genes targeted. The specific mutations of each sample have been shown in Table 2 and S1-S3 figs. The detailed information regarding the mutations in each sample were added to the result section of the manuscript.

“The conversion of Ala � Val at amino acid 168 (A168V) and Thr � Pro at amino acid 127 (T127P) were detected in HBsAg of the occult HBV strain.” This sentence was added to the abstract section.

Page 2, line 36, did the HBcAb positive women have HBsAg or were they only positive for

HBcAb? This is because in your results you state that 2.7% of HBcAb positive were also positive for HBV DNA.

Response: Of 41 HBcAb seropositive women, 4 women were positive for HBsAg (HBsAg positive and HBcAb positive), and 37 women were negative for HBsAg (HBsAg negative and HBcAb positive).

Overall, 26.7% of HBsAg seropositive pregnant women and 2.7% of HBcAb seropositive women had HBV viremia with genotype D, sub-genotype D3, and subtype ayw2:

2.7% (1 case of 37 HBsAg negative and HBcAb positive cases)

26.7% (4 cases of 15 HBsAg positive cases)

Reviewer: Page 13 line 227, can the authors add the percentages on these sentences because they are confusing where they are placed. Then the authors can mention the genotypes without the percentages.

Response: The paragraph was revised.

In page 6 line 128-129 the authors make mention of the act that seropositive women were tested for HIV and HCV, was this part of this manuscript? Or another study? This is because the results section in the abstract makes no mention of these results.

Response: This is a part of this study. Due to limitation of word count in the abstract section, this part had not been added to the abstract section. This part was added to the abstract section.

Page 6 line 132 the authors mention two nested PCR assay however the author only make mention of one PCR assay targeting the S gene.

Response: Two nested PCR assays were described in the manuscript. The first nested PCR assay has been mentioned in the “PCR amplification and sequencing” section. The second nested PCR assay has been mentioned in the “Phylogenetic and mutational analysis” section. This is due to the request of one of the reviewers. Figures 2 and 3 show the electrophoresis of PCR products of these two PCR assays. 

Page 7 line 148-153 falls under the title “PCR amplification and sequencing” from line 153 to 156 should be added to the text under phylogenetic and mutational analysis and rephrased.

Response: The suggestion was applied. This was due to the request of one of the reviewers. 

Page 8, line 177, the authors did not mention, any liver enzyme tests performed however they are mentioning them in their results.

Response: Levels of the liver enzymes were obtained from the records of pregnant women at the public health centers. The pregnant women attend these public health centers for periodical checkups.

Reviewer: can this statement be added in the methods.

Response: This sentence was added to the method section.

Page 8 line 179 after (68.4%) the sentence is unclear and should be rephrased.

Response: It means 68.4%, 55.2%, 97.1% and 90.0% of women were in the third trimester of pregnancy, the age group 20–29 years, educated and Fars, respectively. On the other word, the majority of pregnant women were in the third trimester of pregnancy, the age group 20–29 years, educated, and Fars.

Page 17, line 340, the authors state that one sample was negative for HBeAg however this was not mentioned in the methods that was tested in any sample.

Response: Unfortunately, the presence of HBeAg was not investigated in the serum samples of all participants due to the lack of budget (the available funding could not cover the expenses to test all of the samples). Only the seropositive samples were tested for the presence of HBeAg.

Therefore, detection of HBeAg was not mentioned in the methods section.

Reviewer: it does not matter how many samples were tested but the reason behind the selection of the samples tested should be mentioned otherwise the authors should remove the statement about HBeAg from the results.

Response: The detection of HBeAg was added to the method section.

Page 16 line 368, the authors can according to literature discuss the outcomes of the children infected with the genotypes detected and the implications of the mutations found if they were to be transmitted to the children. I therefor feel that the authors are discussing the results and not discussing the results looking at the population and the implications of the infections transmitted to the children.

Response: Since we could not follow the infants born to infected mothers due to the cross-sectional design of the study, we were not able to determine the eventual effects of hepatitis B on pregnancy outcomes in this study. This part has been mentioned as a limitation of this study. Nevertheless, the overall effects of the detected genotype and mutations in this study have been discussed in the discussion section. These parts were highlighted with yellow color.

“The BCP and pre-C mutations are associated with the progression of chronic HBV infection to advanced liver disease and are frequently found in patients with chronic hepatitis, fibrosis, liver cirrhosis, and HCC.”

“Genotype D is characterized by chronicity, worse clinical outcomes (cirrhosis and HCC), low response to IFN-based therapy, and a high frequency of BCP and pre-C mutations”

Page 17 line 390, the “of HBV-infected women” should come after HBV vaccination so that the recommended can include vaccination of the infected women not just vaccination.

Response: The suggestion was applied.

The last comment is based on vaccinated in Iran yes there is vaccination in infants does the country do a birth dose of vaccine? This should be included in the discussion as a way to reduce the prevalence.

Response: Since 1993, infants receive HBV vaccine at birth in Iran. This has been mentioned at the introduction section. This sentence was highlighted with yellow color. 

“The initiation of HBV vaccination of infants since 1993 and teenagers since 2006 in the country has had a significant role in reducing the prevalence of HBV infection in the community.”

Reviewer #4: The authors have adequately considered the comments made in the preliminary review and revising the manuscript. I propose that the manuscript be reviewed for minor grammatical and typographical errors to enhance clarity and improve readability.

Response: Thank you very much for your review. The manuscript was edited for proper English language, grammar, punctuation, and spelling by our team at Bushehr University of Medical Sciences. The changes have been shown by “Track changes”. 

Sincerely yours,

Fatemeh Farshadpour, Ph.D, 

Bushehr University of Medical Sciences 7514633341 

Bushehr, Iran.

00989171712653

E-mail: f.farshadpour@bpums.ac.ir

f.farshadpour@yahoo.com

---

## [Editor Report · Decision Letter 2]

23 Feb 2022

Prevalence, genotype distribution and mutations of hepatitis B virus and the associated risk factors among pregnant women residing in the northern shores of Persian Gulf, Iran

PONE-D-21-32739R2

Dear Dr.Fatemeh Farshadpour

We’re pleased to inform you that your manuscript has been judged scientifically suitable for publication and will be formally accepted for publication once it meets all outstanding technical requirements.

Kind regards,

Maemu Petronella Gededzha, Ph.D

Academic Editor

PLOS ONE